# Feature engineering and machine learning for computer-assisted screening of children with speech disorders

**Kerul Suthar** [1], **Farnaz Yousefi Zowj** [1], **Marisha Speights Atkins** [2]*, **Q. Peter He** [1]*

**1** Department of Chemical Engineering, Auburn University, Auburn, Alabama, United States of America,
**2** Communication Sciences & Disorders, Northwestern University, Evanston, Illinois, United States of America

☯ These authors contributed equally to this work.
* marisha.speights@northwestern.edu (MSA); qhe@auburn.edu (QPH)

**Data Availability Statement:** All data are in the manuscript and/or supporting information files.

**Funding:** Auburn University Office of the Vice President for Research & Economic Development

## Abstract

Auditory perceptual analysis (APA) is the main method for clinical assessment of speech-language deficits, which are one of the most prevalent childhood disabilities. However, results from APA are susceptible to intra- and inter-rater variabilities. There are also other limitations of manual or hand transcription-based speech disorder diagnostic methods. There is increased interest in developing automated methods that quantify speech patterns for diagnosing speech disorders in children to address these limitations. Landmark (LM) analysis is an approach that characterizes acoustic events occurring due to sufficiently precise articulatory movements. This work investigates the utilization of LMs for automatic speech disorder detection in children. Besides the LM-based features that have been proposed in existing research, we propose a set of novel knowledge-based features that have not been proposed before. A systematic study and comparison of different linear and nonlinear machine learning classification techniques based on the raw features and the proposed features is conducted to assess the effectiveness of the novel features in classifying speech disorder patients from normal speakers.

## 1 Introduction

Speech-language deficits are one of the most prevalent childhood disabilities affecting about 1 in 12 children between three and five years old [1]. Despite the recognition that early identification and treatment of communication disorders is important for school readiness and has been shown to significantly improve communication, literacy, and mental health outcomes for young children [1–3], approximately 40% of children with speech and language disorders do not receive intervention because their impairment goes undetected [4,5]. Auditory perceptual analysis (APA) is the main method for clinical assessment of disordered speech; however, results from APA are susceptible to intra- and inter-rater variabilities [6]. Another factor to consider is that some children may be reluctant to participate in long testing sessions [7], and even if they do, transcription of large data sets of audio recordings is time-consuming and requires a high level of expertise from therapists [8,9]. These limitations of manual or hand

Innovation Award (MSA). The funders had no role in study design, data collection and analysis, decision to publish, or preparation of the manuscript.

**Competing interests:** The authors have declared that no competing interests exist.

transcription based diagnostic assessment methods have led to an increasing need for automated methods to quickly and consistently quantify child speech patterns and help them be diagnosed if they have impaired speech [10]. Landmark (LM) analysis is such an approach that characterizes speech with acoustic markers that are developed based on the LM theory of speech perception [11–13]. Unlike automatic speech recognition (ASR), LM analysis does not attempt to identify words, but rather to detect acoustic events that occur as the result of sufficiently precise articulatory movements. LM analysis has been suggested as the basis for automatic speech analysis [6]. Therefore, in this work we focus on the utilization of LMs for automatic speech disorder detection in children, with LMs extracted using a publicly available software: SpeechMark toolbox [6]. SpeechMark not only analyzes physical aspects of the signal but also applies acoustic knowledge of articulatory features in the process of analysis [6]. As a result, it has been utilized in numerous studies to extract LMs for various applications such as the detection of stress [14], depression [15], emotion [16], and sleep deprivation [17]. Here we briefly review how it works. SpeechMark divides the computed spectrogram into six frequency bands, and then fine and coarse processing steps are conducted to detect band-energy rise and to determine threshold for peak detection. Finally, energy peaks are located and LM types are determined based on the patterns of changes in the frequency bands [12,13]. The description of each landmark detected by this tool and used in this study are presented in Table 1. An example of LMs detected by SpeechMark from the speech of a speaker uttering a word are shown are shown in Fig 1.

An extension of SpeechMark for the landmark system, namely automatic syllabic cluster analysis, was recently proposed that clusters the LMs into syllabic units [18]. These LMs are grouped based on specific rules, such as at least a 30 ms voiced segment is required in a syllabic cluster (SC) [19]. Studies show that SC patterns are good indicators of differences between normal and disordered speakers [18,19]. Variations in articulatory exactness in normal and disordered speakers have proven to be related to LM and SC patterns [19]. This is validated by existing studies showing that simple count of LMs and/or SCs can be used for classification of gender [6], Parkinson's disease [20], and sleep deprivation [17]. Counting of individual LMs, a.k.a. unigrams, does not consider the specific order or sequence of the LMs, which may contain important information about the speech. $n$-gram, which is a generalization of unigrams and is defined as a sequence of $n$ consecutive LMs, takes the specific LM order into consideration when $n \geq 2$ [21]. It was found that $n$-gram counts ($n = 1,2,3,4$) were good features for depression detection [15,22]. For example, in [22], SpeechMark was used to extract landmarks from a large dataset consisting of recordings from smartphones. Two sets of features were proposed based on speech landmark bigrams, *i.e.*, bigram-count and LDA-bigram. The first set calculates the frequencies of bigrams, and the second set detects latent patterns from bigrams using natural language text processing. A linear support vector machine (SVM) classifier was trained using the two sets of features. It was found that the bigram features increased the accuracy of the SVM classifier from 72.9% when only acoustic features were used to 78.7% when either bigram-count or LDA-bigrams were utilized. The speech landmark bigram features

**Table 1. Description of landmarks used in this study.**

| Landmark | Description |
| --- | --- |
| g (glottis) | Onset (+) and offset (-) of sustained motion of vocal fold |
| b (burst) | Onset (+) and offset (-) of frication or bursts in an unvoiced segment |
| s (syllabicity) | Release (+) and closure (-) of sonorant consonant in a voiced segment |
| f (unvoiced frication) | Onset (+) and offset (-) of frication in an unvoiced segment |
| v (voiced frication) | Onset (+) and offset (-) of frication of in a voiced segment |

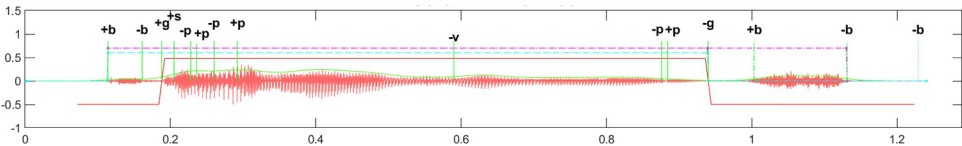

**Fig 1. Detected landmarks (LMs) using SpeechMark.**

improved the F1(depressed) by 30.1% compared to acoustic features [22]. Besides *n*-gram count, time-based LM features have also been proposed in the literature. These time-based LM features include durations of the bigrams (*i.e.*, 2-grams) and LM pairs (*i.e.*, onset and offset of a LM as defined in Table 1) [15], and speech rate, which is defined as the number of phonetic units, such as syllables or words, uttered per unit time [23,24].

In this work, we have adopted count of *n*-grams as well as duration and rate features based on LMs and n-grams. One contribution of this work is to propose novel knowledge-based features that have not been proposed before and to demonstrate the effectiveness of these new features in detecting childhood speech disorder. For example, this work studies features that are the ratios of the count of *n*-grams ($n \geq 2$) to that of unigrams. The idea of considering ratio is similar to the body-mass index (BMI) where the weight itself cannot determine whether a person is overweight or not. BMI takes the height of the person into account as well. The ratios are usually better features than the absolute individual values in addressing the individual variations of samples within the same class. This point is further validated in this study. Another contribution is to perform systematic feature selection to identify key features and quantify their contributions to the classification of patients with speech disorder from normal controls. The final contribution of this work is a systematic study and comparison of different linear and nonlinear machine learning classification techniques and their effectiveness in classifying speech disorder patients from normal speakers.

The remainder of this work is organized as follows. Section 2 describes materials used in this study, which include the general information about the speakers of which the speech samples were collected, the conditions and procedures the speech samples were processed to obtain the dataset, and the features proposed to be studied in this work. Section 2 also introduces the analytical methods used in this study, which include methods to address the data imbalance, introduction of machine learning (ML) classification techniques used in this work, and the procedure and criteria used to evaluate the performance of different ML techniques in screening children with speech disorders. Section 3 presents results and discussions of this work, and Section 4 draws some conclusions.

## 2 Materials and methods

### 2.1 Ethical considerations

Ethical approval for human subject data collection was granted by the University of Cincinnati for this study with reference number 2015–3023 and subsequently at Auburn University with reference number 17–203 EP 1705 for ongoing data analysis and additional data collection. Permissions were also sought from schools and university clinics where data were collected. Anonymity and confidentiality were explained to participants. Participants were assured that withdrawal from study would not harm them in any way. Informed consent forms were filled and signed by parents with verbal assent from the child participants. Participant's data was de-identified with codes to ensure anonymity. All data analyses in this work are conducted using the de-identified data.

## 2.2 Speakers

The speech of 52 children ages 33–94 months (with mean 51.52 and standard deviation 10.16) was retrieved from the Speech Evaluation and Exemplars Database (SEED) [25]. Due to missing values, one sample was dropped from this work. Of the 51 remaining children, 39 were typically developing without speech or language disorder, and 12 were diagnosed with speech sound disorder without language impairment. All children were required to demonstrate normal hearing using the criterion of sound detection at 20 dB HL for pure tones at 500, 1000, 2000, and 4000 Hz. Participants were required to exhibit age-appropriate receptive language skills on the CELF Preschool-2 [26]. Age-appropriate performance was determined by scores falling within one standard deviation of the mean (standard score > 85). Children were classified as typically developing or with speech disorder using the Clinical Assessment of Articulation and Phonology-2 or the Diagnostic Evaluation of Articulation [27]. Children with standard scores ≤ 85 (one standard deviation below the mean) were assigned to the with speech disorder group. Children with concomitant language disorders were not included in the study.

## 2.3 Dataset

The speech samples retrieved were recorded in local community early education centers or in the lab. Sound levels were measured prior to each recording session to determine if the environmental noise level was below 40 dBA SPL in both the school and lab environment (Williams, Zhou, Stewart, & Knott, 2016). Speech samples were recorded at a 44K sampling rate at 24-bit depth using a handheld ZOOM H6N recorder (Zoom North America) with cardioid XLR MOVO LV402 microphones (MOVO). Speech samples retrieved for this study were one of the Triage 10 word set from Anderson and Cohen [28]: *flower*. Acoustic landmarks, including +/-g, +/-b, +/-s, +/-f, and +/-v, as well as syllabic clusters were obtained using the SpeechMark MATLAB toolbox (STAR Corp., MA).

## 2.4 Feature engineering

The raw features extracted from audio recordings using the SpeechMark Toolbox include time stamp and strength of each LM listed in Table 1, plus SC count. As discussed previously in Sec. 1, in this work we have adopted all LM and SC based features proposed in the literature, including *n*-gram counts, and duration and rate features based on LMs and *n*-grams. These features are listed in the top rows of Table 2. In addition, we explore LM strength based features and propose *n*-gram ratio based features to better address within-class variations as discussed in Sec. 1. These new features are listed in the bottom rows of Table 2. After removing illegitimate or trivial features (e.g., n-gram counts that are all zeros, or ratios with a denominator of zero), there are 303 unique features generated based on the criteria listed in Table 2.

## 2.5 Feature selection

It has been shown by many studies that the performances of classification methods can be significantly improved if only the relevant features are included as the predictors. Feature selection can also reduce the risk of overfitting, which is especially important when the number of samples are relatively small compared to the number of features (such as the case of this study). Finally, feature selection can reduce model complexity, making result interpretation easier. As a result, feature selection has been one of the most important practical concerns in data-driven approaches. In the past few decades, many different feature selection approaches have been reported for various modeling and classification applications. For more detailed

**Table 2. Features employed in this study–including features adopted from literature and features proposed in this work.**

| Feature category | Description | Unit |
|---|---|---|
| **Features adopted from literature** | | |
| Unigram count | Number of each unigram type | # |
| Bigram count | Number of each bigram type | # |
| Trigram count | Number of each trigram type | # |
| Average bigram duration | Average duration of all bigrams | s |
| Average trigram duration | Average duration of all trigrams | s |
| Duration of LM pair | Average duration of LM pairs (*i.e.*, onset-offset) of each LM type | s |
| Unigram rate | Count of all unigram types per unit time | #/s |
| Bigram rate | Count of all bigram types per unit time | #/s |
| Trigram rate | Count of all trigram types per unit time | #/s |
| Syllabic cluster count | Number of syllabic clusters | # |
| Speech rate | Syllabic cluster count per unit time | #/s |
| **New features proposed in this work** | | |
| Strength of unigram | Average strength of each unigram type | % |
| Strength of bigram | Average strength of each bigram type | % |
| Strength of trigram | Average strength of each trigram type | % |
| Strength change | Average absolute strength difference of two consecutive LMs | % |
| Average bigram strength | Average strength of all bigram types | % |
| Average trigram strength | Average strength of all trigram types | % |
| Unigram/unigram ratio | Ratio of unigram counts of each type | - |
| Bigram/unigram ratio | Ratio of bigram count to unigram count of each type | - |
| Trigram/unigram ratio | Ratio of trigram count to unigram count of each type | - |

discussions on various feature selection methods, the readers are referred to some recent review articles.

In this work, a two-step feature selection procedure is proposed. In the first step, the redundant features (*i.e.*, the features that are highly correlated with an existing feature) are removed. In the study, a Pearson correlation coefficient of 0.99 is used as the criterion to determine whether a feature is redundant with an existing feature or not. After this step, the number of features is reduced to 189 from the original 303 features, indicating that there is significant redundancy among the original features.

In the second step, the recursive feature elimination with cross-validation (RFECV) from *scikit-learn* is utilized with the default 5-fold cross-validation. Fig 2 shows the cross-validation score vs. number of features when a linear discriminant analysis (LDA) model is used as the classifier. Fig 2 indicates that only 10 features are needed to obtain the optimal cross-validation score. The 10 features selected are listed in Table 3. As can be seen from Table 3, nine out of the ten features are new features proposed in this work that have not been utilized before. Among the nine new features, seven are ratio-based features and two are strength-based features.

## 2.6 Sample imbalance

Among all 51 samples, 39 samples belong to normal speakers, while the remaining 12 samples belong to the disordered speakers, which indicates an approximately 3:1 class imbalance between the normal speaker samples and the disordered ones. However, most machine learning classification algorithms are developed with the implicit assumption of approximately

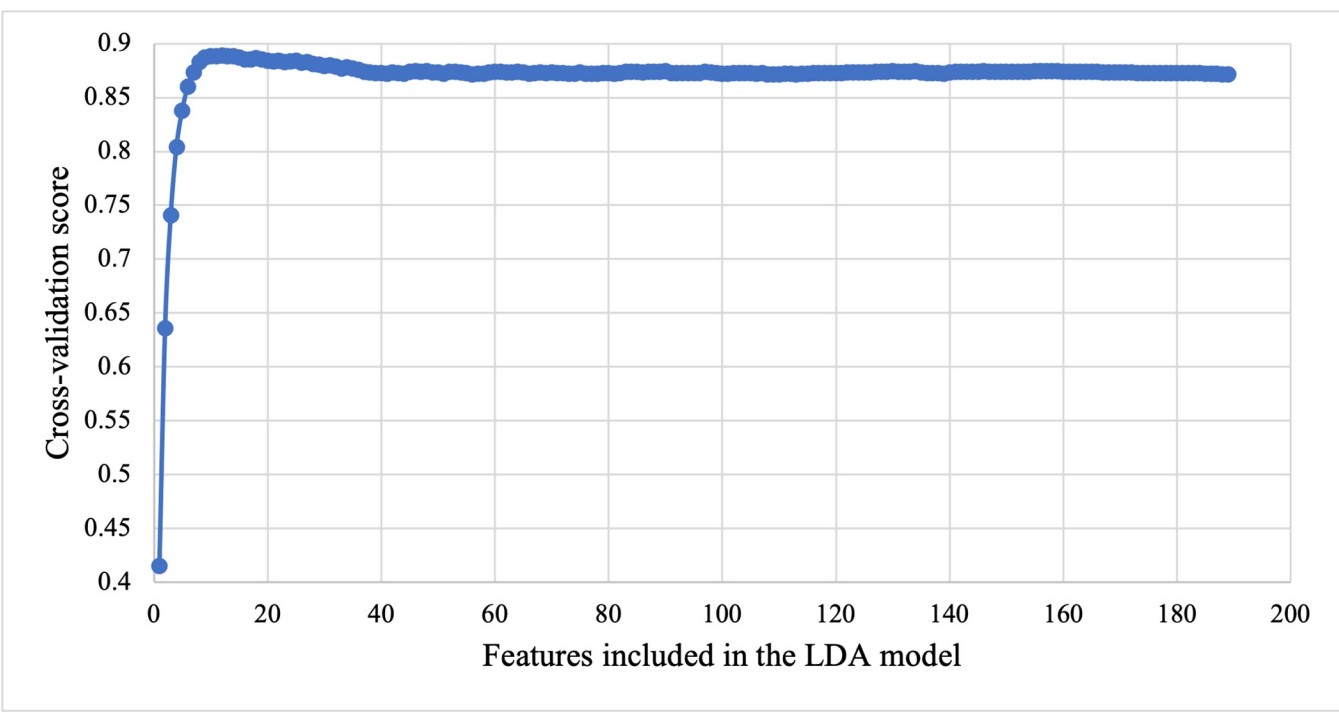

**Fig 2. Recursive feature elimination with cross-validation (RFECV).**

equal samples in each class. Therefore, data with imbalanced or skewed classes may result in poor classification performance for the minority class samples. For example, if the model is tuned using accuracy, the resulted model may lead to mostly correct classification of the majority class at the cost of poor classification of the minority class. However, the correct classification of minority class samples is often more critical as they represent the disease group most of the time–misclassification of these samples leads to low sensitivity. Several ways of dealing with class imbalance have been proposed in the literature such as under-sampling, over-sampling, synthetic sample generation, using cost-sensitive methods, and applying penalties or weights based on class ratio [29]. Under-sampling reduces the number of samples in the majority class to improve the imbalance ratio, while oversampling refers to increasing the number of samples in the minority class samples. Oversampling is used more often than

**Table 3. The ten features selected based on RFECV.**

| Feature category | Feature specifics |
| --- | --- |
| Ratios of bigram count to unigram count | '-g-b/+g' |
| Ratios of trigram count to unigram count | '-s+s-s/+g' |
| Ratios of trigram count to unigram count | '+b+g+s/+g' |
| Ratios of bigram count to unigram count | '-b+g/+g' |
| Trigram counts | '+b-b+b' |
| Ratios of trigram count to unigram count | '-s-g+b/+g' |
| Ratios of trigram count to unigram count | '-g+b-b/+g' |
| Strength of trigrams | '+g-g-b' |
| Strength of unigrams | '-f' |
| Ratios of trigram count to unigram count | '+b+g-v/+g' |

under-sampling to maximumly utilize available samples. Random oversampling refers to an increase in minority class samples through duplication of randomly selected minority class samples. However, this most straightforward approach that duplicates the existing samples does not add new information during training and is not considered robust. A more robust method when oversampling a dataset is the synthetic minority over-sampling technique (SMOTE), in which new samples are synthesized from the existing samples [30]. SMOTE is utilized in this work to address the class imbalance issue and we briefly review the technique and the implementation details in the following subsection.

## 2.7 Synthetic minority over-sampling technique (SMOTE)

Based on the feature space of the minority samples, SMOTE first selects a minority class instance or sample at random (denoted as *a*) and then finds its *k* nearest minority class neighbors. The synthetic neighbor is created by selecting one of the *k* nearest neighbors at random (denoted as *b*) and connecting them to form a line segment in the feature space. The synthetic sample is generated as a combination or linear interpolation between the two chosen samples, *a* and *b*, as follows.

$$\boldsymbol{x}_{new} = \boldsymbol{x}_a + \lambda \left( \boldsymbol{x}_b - \boldsymbol{x}_a \right) \tag{1}$$

where $\boldsymbol{x}_i$ denotes the feature vector (*i.e.*, a point in the feature space) of sample $i$, $\lambda$ is a random number in the range [0, 1]. For features that only take integer values (e.g., *n*-gram counts), $\boldsymbol{x}_{new}$ is rounded to the nearest integer. More information on SMOTE can be found in [30]. For implementation, Python-based library *imbalanced-learn* was used in our work for SMOTE oversampling [31]. Due to limited data, we first randomly isolate one sample from each class for testing. Once the two random samples, *i.e.*, one from a normal speaker and one from a disordered speaker, are removed from the set, we apply SMOTE oversampling to balance the dataset. The primary purpose behind separating test samples before oversampling is to avoid bias in the model due to the test samples' influence on the synthetic samples created. After removing one sample from each class for testing, we have 38 samples from normal speakers and 11 samples from disordered speakers in the training set. SMOTE oversampling is applied on the training set, where 27 samples in the minority class, i.e., disordered speakers, are generated to balance the dataset.

## 2.8 Monte-Carlo cross validation and testing (MCVT)

Once the training set is balanced, we train different classification models, perform feature selection and tune their hyperparameters using 10-fold cross-validation on the training set. Then we apply the models to the left-out test samples, and report the sensitivity and specificity of each model. This whole procedure is referred to as one Monte-Carlo validation and testing (MCVT) [32]. We report the mean and standard deviation of sensitivity and specificity of 50 such MCVT runs, which is a robust way of comparing different modeling techniques and assessing their performances. MCVT avoids overfitting by randomly selecting and isolating two test samples first, then utilizing SMOTE technique to balance the rest of the dataset, which is further split into training and validation for modeling training, feature selection, and hyperparameter tuning. For hyperparameter tuning, we use a 10-fold stratified cross-validation (CV) to select the optimal hyperparameters. The schematic of the proposed MCVT procedure is shown in Fig 3.

Sensitivity and specificity are two most commonly used critical metrics when dealing with binary classification problems in healthcare. Sensitivity is the true positive rate, i.e., the classifier's ability to detect diseased patients correctly, and specificity is the true negative rate, i.e., the

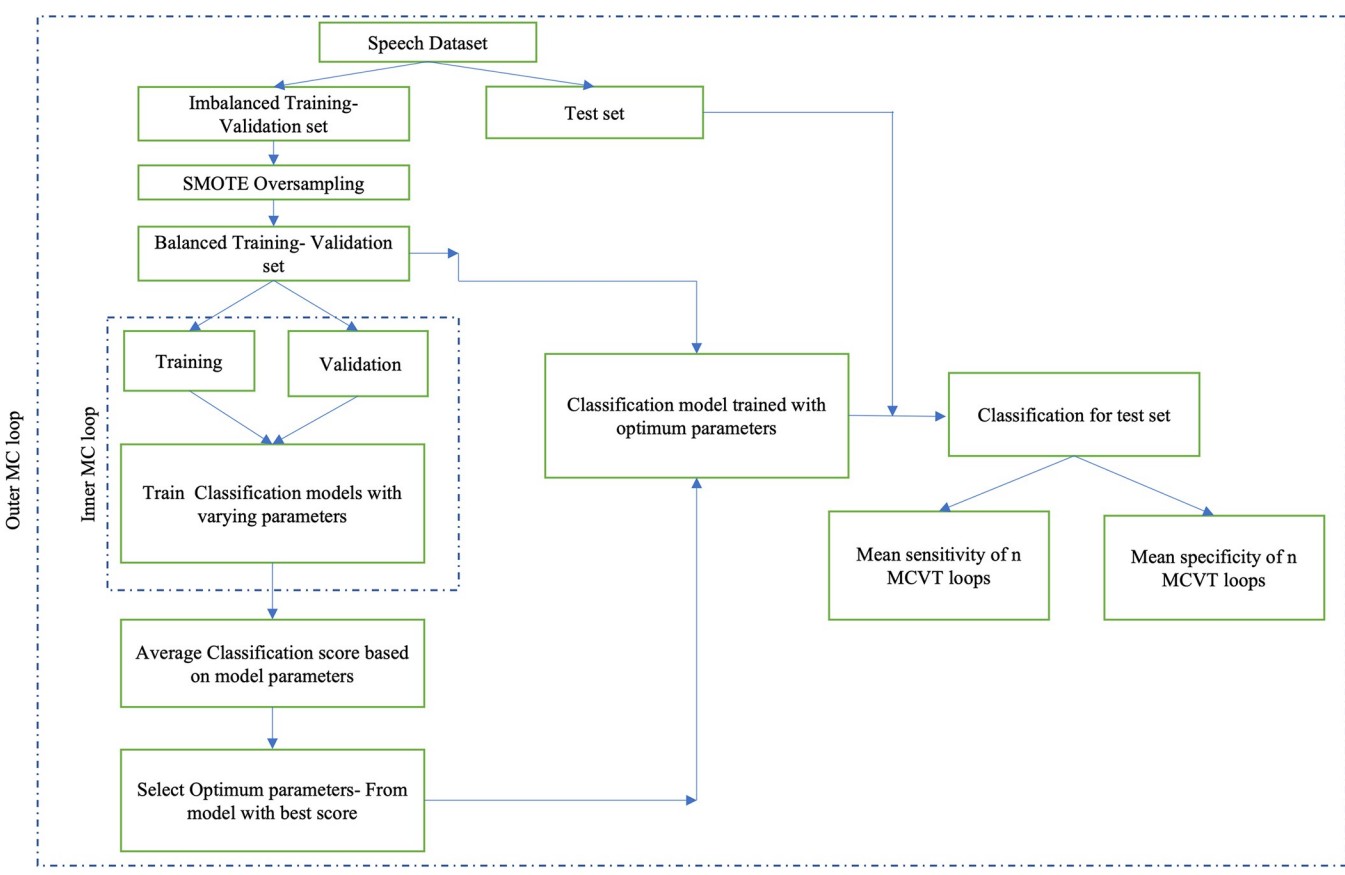

**Fig 3. Schematic of the MCVT for comparing different modeling techniques and assessing their performances in terms of accuracy and robustness.**

classifier's ability to detect normal controls (i.e., the ones without diseases) correctly. We also use accuracy as a single measure when we need to evaluate the overall performance of a classifier. The mathematical definitions of these terms are given below.

$$Sensitivity = \frac{n_{TP}}{n_{TP} + n_{FN}} \tag{2}$$

$$Specficity = \frac{n_{TN}}{n_{TN} + n_{FP}} \tag{3}$$

$$Accuracy = \frac{n_{TN} + n_{TP}}{n_{TN} + n_{TP} + n_{FN} + n_{FP}} \tag{4}$$

where $n_{TP}$ is the number of true positives, $n_{FN}$ number of false negatives, $n_{TN}$ number of true negatives, and $n_{FP}$ number of false negatives. Sensitivity, specificity and accuracy all range from 0 to 1 (or 0~100%).

As shown in Fig 3, the mean or average sensitivity and specificity of MCVT runs can be used to assess the accuracy of a classifier; while the standard deviations of the sensitivity and specificity of the MCVT runs can be used to quantify the robustness of a classifier (i.e., how consistently a classifier performs when trained with randomly selected training samples). It is worth noting that because only one sample from each class is left out for testing in this work,

the standard deviations of the sensitivity and specificity would be biased due to the extremely small sample size. Therefore, this measure of robustness is not utilized in this work.

### 2.9 The trade-off between sensitivity and specificity

For binary classification, there is often a trade-off between sensitivity and specificity. This trade-off can be visualized in a receiver operator characteristic (ROC) curve, which plots sensitivity vs. (1 –specificity) as illustrated in Fig 4. The area under the curve (AUC) is the summative measure of the classification capability of a classifier. A perfect classifier has an AUC of 1, while a classifier with random selection has an AUC of 0.5. In reality, a typical classifier has AUC between 0.5 and 1. In selecting operation points on ROC, often times the costs associated with misclassification of each type must be considered when trying to balance the sensitivity with the specificity. This balance is usually adjusted through class priors or class weights. For example, to use the proposed method as a screening tool, we may want to trade (or sacrifice) some specificity for higher sensitivity. This is because if a truly disordered speaker were misclassified as a normal speaker, he or she may miss the opportunity to be further examined by a speech specialist.

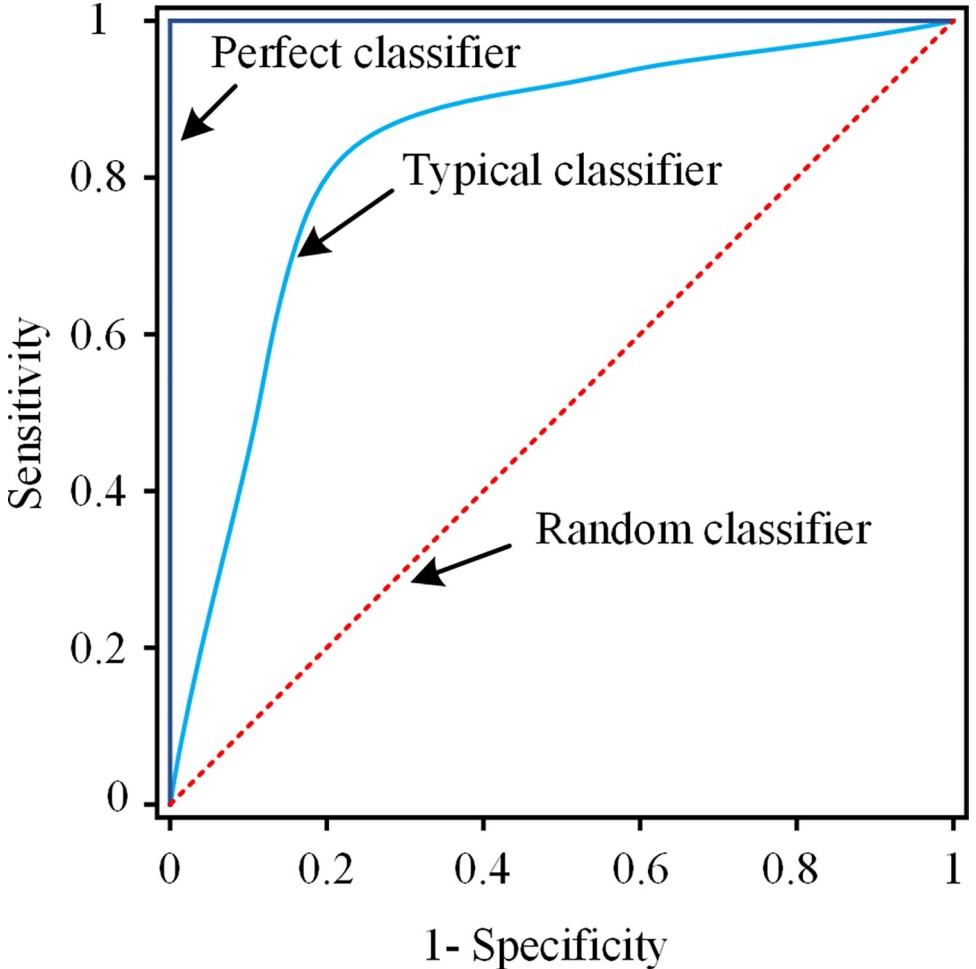

**Fig 4. The trade-off between sensitivity and specificity illustrated by a receiver operator characteristic (ROC) curve.**

## 2.10 Classification techniques

In this work, four different classification algorithms, namely linear discriminant analysis (LDA), support vector machine (SVM), extreme gradient boosting (XGBoost), and random forest (RF). For LDA, we consider the effects of shrinkage, a form of regularization to avoid overfitting, along with class priors, to address the unequal costs of misclassification. For SVM, we examine the effects of different kernels, along with class weights. Throughout the modeling procedure, grid search and random search are used for hyperparameter tuning using the scikit-learn library in Python [33]. However, class weights or priors are not tuned automatically, and certain discrete values are considered for the study. We consider several categories based on features extracted using feature engineering and different levels of feature selection. We compare raw data with different feature categories.

**2.10.1 Linear discriminant analysis.**   Linear discriminant analysis (LDA) is one of the most commonly used linear classification techniques used in machine learning. In this work, the LDA function from Python *scikit-learn* library [34] is used, which generates the linear decision boundary based on the Bayes' rule by modeling the class conditional or posterior probability $P(y = k|x_i)$ of each training sample of $d$ features (*i.e.*, $x_i \epsilon R^d$) for each class $k$:

$$P(y_i = k|x_i) = \frac{P(x_i|y_i = k)P(y_i = k)}{P(x_i)} = \frac{P(x_i|y_i = k)P(y_i = k)}{\sum_l P(x_i|y_i = l)P(y_i = l)} \tag{5}$$

where $y$ is the class label and the class $k$ that maximizes the posterior probability is selected.

Shrinkage is a form of regularization used in LDA to improve the estimation of covariance matrices in situations where the number of training samples is small compared to the number of features. The effect of shrinkage is studied in this work. In addition, the prior probability, $P(y = k)$, is studied on its effectiveness in addressing the unequal costs of misclassification. More information can be found in [34].

**2.10.2 Support vector machine.**   Support vector machine (SVM) is a classification approach developed in the 1990s. Various SVMs have shown superior performance in a variety of settings and are often considered one of the best "out of the box" classifiers [35]. For simple interpretation and to reduce the risk of overfitting, in this work we focus on two-class linear SVM. Consider $n$ samples each with $d$ features (*i.e.*, $x_i \epsilon R^d$, $i = 1, \ldots, n$) and their labels $y_i \epsilon \{+1, -1\}$, linear SVM identifies a hyperplane, which is a liner function in the feature space, i.e., $f(x) = \langle w, x \rangle + b$, where $w$ is the coefficient vector, $b$ is a real constant, and $\langle \cdot, \cdot \rangle$ denotes the dot production in the feature space. The hyperplane is placed such that a maximum distance between the two class samples (i.e., the class margin) is achieved. This is equivalent to the following minimization problem [36]:

$$\min_{w,b} \frac{1}{2} w^T w \tag{6}$$

$$s.t.\ y_i(< w, x_i > +b) - 1 \geq 0, \forall i \tag{7}$$

For a nonseparable case, a soft margin is introduced so that the minimization problem becomes

$$\min_{w,b,\xi} \frac{1}{2} \|w\|^2 + C\sum_{i=1}^{n} \xi_i \tag{8}$$

$$s.t.\ y_i(< w, x_i > +b) \geq 1 - \xi_i, \xi_i \geq 0, i = 1, \ldots, n \tag{9}$$

where $C$ and $\xi_i$ are constants and $\xi_i$ are called slack variables.

More information on VSM and its training can be found elsewhere [35,36]. In this work, *Scikit-learn* [33] with LIBSVM [37] library is used to implement linear SVM. For comparison, we also implemented SVM with nonlinear kernels, including polynomial, radial basis function (RBF), and sigmoid kernels.

**2.10.3 Random forest.** Random forest (RF) is an ensemble method of decision tree algorithms. It is an extension of bootstrap aggregation or bagging of decision trees. In bagging, each classifier's training set is generated by random sampling, with or without replacement from all the samples available for training. Individual predictions of each classifier are aggregated based on a hard or soft voting scheme to form a final prediction. However, unlike bagging, RF also involves selecting a subset of input features at each split point in the construction of trees. *Scikit-learn* is used for implementation of RF. Hyperparameters, including number of trees, max tree depth, and number of features considered for the split, are tuned using random search hyper parameter optimization procedure. More information on RF can be found in [35,38].

**2.10.4 Extreme gradient boosting (XGBoost).** Another decision tree-based ensemble method used in this work is boosting. In comparison to bagging, boosting approaches combine various homogenous weak learners and learn patterns sequentially in an adaptive way. Each of the sequential model depends on the previous ones. XGBoost is one of the most popular boosting approaches, which has been used widely and has achieved state-of-the-art results on many machine learning challenges [39]. XGBoost is an optimized distributed gradient boosting library, which is implemented under the gradient boosting framework. More information on XGBoost can be found in [39].

## Results and discussion

In this work, we conduct investigation from two perspectives: (1) comparing classification performance when different feature sets are used, and (2) comparing classification performance when different classification techniques are used. When comparing different features, the following three feature sets are studied: (a) the original 21 features directly obtained from the SpeechMark Toolbox, which include the counts and strengths of the ten LMs (listed in Table 1, considering both onset and offset) for each sample, plus one syllabic count per sample; (b) the 189 features based on rational feature engineering with different feature types listed in Table 2 and after redundant features (*i.e.*, Pearson correlation coefficient greater than or equal to 0.99) removed; and (c) The ten features selected from the 189 features via RFECV as discussed in Sec. 2.5. These ten features are listed in Table 3. When comparing different classification techniques, they are applied to all the three feature sets.

It is worth noting that all the results presented in this work are based on the unseen test data (*i.e.*, they are not involved in any training steps such as feature selection or hyper-parameter tuning). Due to the small number of samples, we would not have enough data for model training if 20~30% of the dataset were left out for testing as usually recommended. As a result, we leave one sample out from each class for the test set (*i.e.*, totally 2 samples in the testing set). To avoid bias or cherry-picking due to the small number of the test samples, we perform 50 Monte Carlo cross-validation and testing (MCVT) and use the average of the 50 MCVT runs for performance evaluation. As we have demonstrated previously, this method provides robust and fair evaluations even with small number of samples [32].

As shown in Table 4 and Fig 5, when the 21 raw features are used, SVM with RBF kernel provides the best overall classification performance with 75.0% accuracy (i.e., 75.0% of the samples are classified correctly). SVM with linear kernel provides the second-best result with 71.0% accuracy. The overall performances of all methods, linear or nonlinear, are relatively poor, indicating that the raw features are not very informative in classifying the two classes.

**Table 4.  Comparison of classification performance based on raw features.**

| Method | Sensitivity (%) | Specificity (%) | Accuracy (%) |
|---|---|---|---|
| LDA | 64 | 54 | 59 |
| SVM (Linear) | 68 | 74 | 71 |
| SVM (Poly) | 78 | 32 | 55 |
| SVM (RBF) | 70 | 80 | 75 |
| SVM (Sigmoid) | 80 | 54 | 67 |
| XGBoost | 50 | 86 | 68 |
| RF | 28 | 76 | 52 |

Next, we apply different classification methods to the selected ten features obtained through rational feature engineering and selection. The results are listed in Table 5 and shown in Fig 6. By comparing Tables 4 and 5 (or Figs 5 and 6), we can see that the performances of all methods have improved in terms of sensitivity, specificity and accuracy. While some improvements are moderate, such as those of SVM (RBF) and XGBoost (with less than 10% improvement in accuracy), others are significant (with as high as 34% improvement in accuracy). Recall that out of the ten selected features, nine of them are newly proposed features. The notable improved performance with these features across all classification methods demonstrates that the proposed features are more informative than the raw features. Sincere there are seven features that are ratio based, the improved performance is most likely due to our hypothesis that ratio-based features are better at addressing individual variations of samples from the same class. The direct comparison of accuracy using the two sets of features are shown in Fig 7. In particular, LDA classifier achieves 94.0%, 92.0% and 93.0% in sensitivity, specificity and overall accuracy respectively. Several other methods also achieve nearly 90.0% in sensitivity, specificity and overall accuracy, including SVM with linear, polynomial and sigmoid kernels. In addition, using raw features has led to skewed or imbalanced sensitivity and specificity in several methods as shown in Fig 5. For example, SVM with polynomial and sigmoid kernels have high

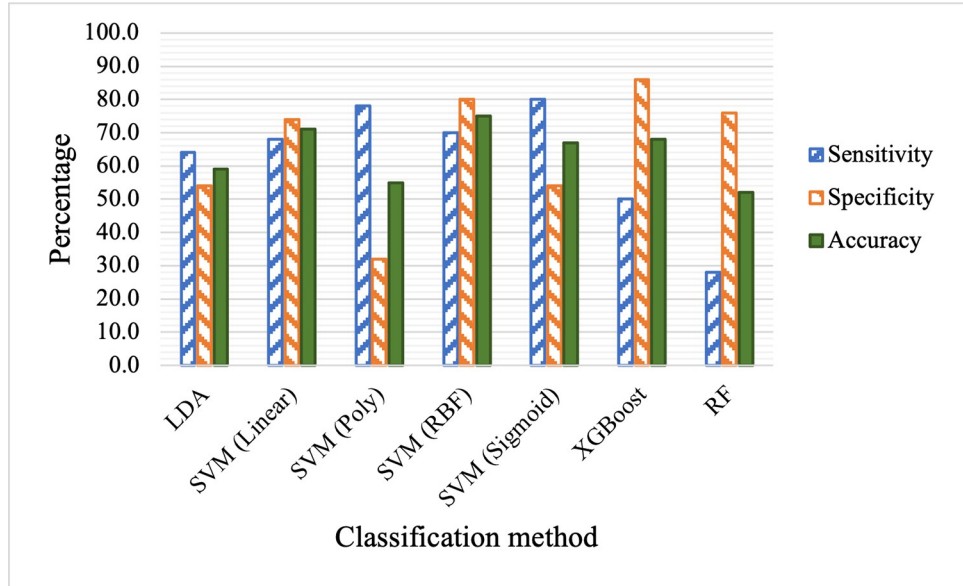

**Fig 5. Comparison of classification performance based on raw features.**

**Table 5. Comparison of classification performance based on rationally engineered and selected features.**

| Method | Sensitivity (%) | Specificity (%) | Accuracy (%) |
|---|---|---|---|
| LDA | 94 | 92 | 93 |
| SVM (Linear) | 86 | 92 | 89 |
| SVM (Poly) | 84 | 94 | 89 |
| SVM (RBF) | 72 | 94 | 83 |
| SVM (Sigmoid) | 88 | 88 | 88 |
| XGBoost | 60 | 88 | 74 |
| RF | 76 | 94 | 85 |

sensitivity but poor specificity, while XGBoost and RF have high specificity but poor sensitivity. In comparison, the sensitivity and specificity based on the selected engineered features are much more balanced.

As discussed previously, class prior probability or class weight can have an impact on sensitivity and specificity, which can also be adjusted to count for the unequal costs of misclassification (i.e., the cost of false positive vs. that of false negative). In this work, since we aim to develop a screening method, high sensitivity is more desirable as the false negative (i.e., children with speech disorder are misclassified as normal speakers) may miss the opportunity to be examined by a speech specialist. On the other hand, false positives will cause less harm other than the cost associated with the follow up examination. Since LDA performs the best among all methods and it is more robust than some of the other nonlinear methods, we focus on examining the impact of class priors on LDA. Three different class priors are studied, namely (0.5,05), (0.3, 07), and (0.1, 0.9). (0.5,0.5) indicates equal priors for normal and disordered classes. (0.1, 0.9) indicates eight times higher prior probability for the disordered class than the normal control group. The expectation is that the priors of (0.1, 0.9) would lead to higher sensitivity compared to (0.5, 0.5) or (0.3, 0.7). The results are shown in Table 6 and Fig 7, which indicate that the sensitivity and specificity are not significantly affected by the class priors. Specifically, there is no change in sensitivity and specificity when class priors are

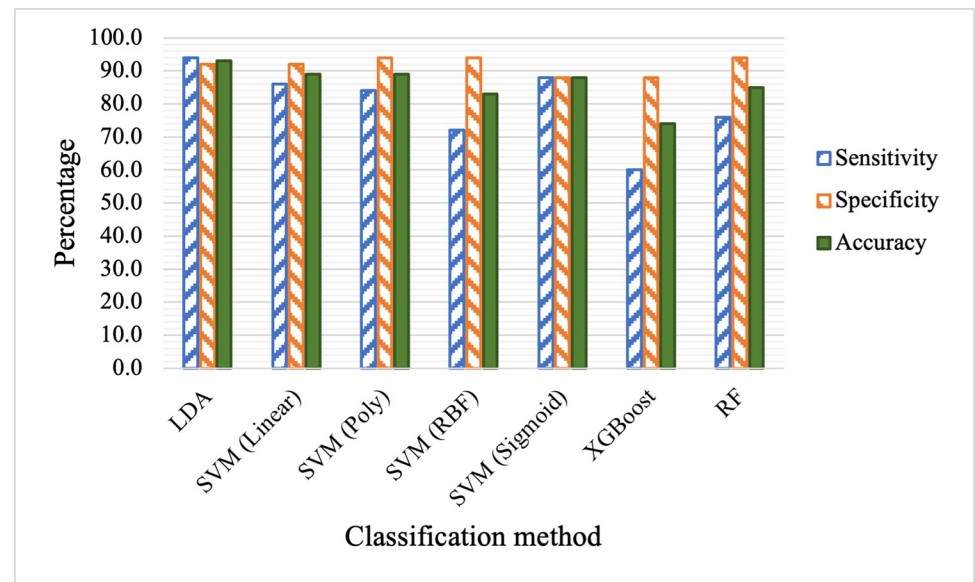

**Fig 6. Comparison of classification performance when selected features are used.**

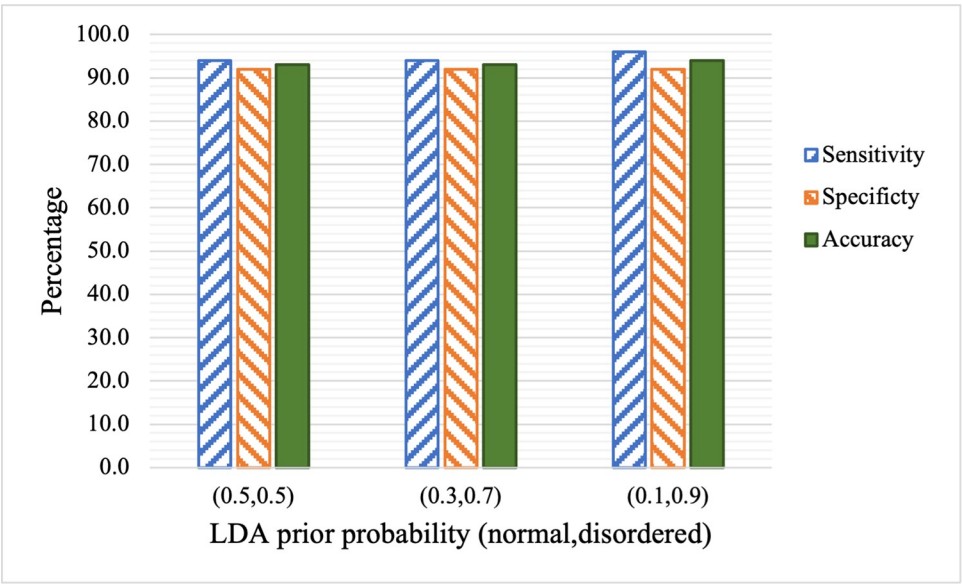

**Fig 7. The impact of class priors on sensitivity, specificity and accuracy of the LDA classifier.**

changed from (0.5,0.5) to (0.3, 0.7). There is slight increase in sensitivity (from 94.0% to 96.0%) when class priors of (0.1, 0.9) are used, while specificity is unchanged.

More thorough examination of the trade-off between sensitivity and specificity the LDA classifier is shown in the ROC curve (Fig 8), which is obtained by changing class priors in a much wider range than the three cases presented previously. As discussed previously, the best possible classification method would yield a point in the upper left corner of the ROC space, representing 100% sensitivity and 100% specificity. As shown in Fig 8, with proper tuning, the ROC curve of LDA approaches that point with 96% sensitivity and 92% specificity. The ROC curve can serve as a visual tool for selecting LDA tuning parameters, i.e., class priors, based on cost/benefit analysis of the speech disorder screening decision making.

## 4. Conclusion

In this work, we propose an automated computer-assisted screening method for children with speech disorders. The main contribution of this work is to propose a set of novel knowledge-based features that have not been proposed before and to demonstrate the effectiveness of these new features in detecting childhood disorder. In particular, this work proposes specific and average strength of n-grams and ratio-based features. The ratio-based features have been found particularly informative in characterizing audio recordings for speech disorder detection. Similar to the idea of BMI metric used in obesity studies, the ratio-based features proposed in this work are hypothesized to better address the usually wide individual variations among samples from the same class than their individual components. This is validated by the

**Table 6. LDA classification performance when different class priors are used.**

| Prior | Sensitivity | Specificity | Accuracy |
|---|---|---|---|
| (0.5,0.5) | 94.0 | 92.0 | 93.0 |
| (0.3,0.7) | 94.0 | 92.0 | 93.0 |
| (0.1,0.9) | 96.0 | 92.0 | 94.0 |

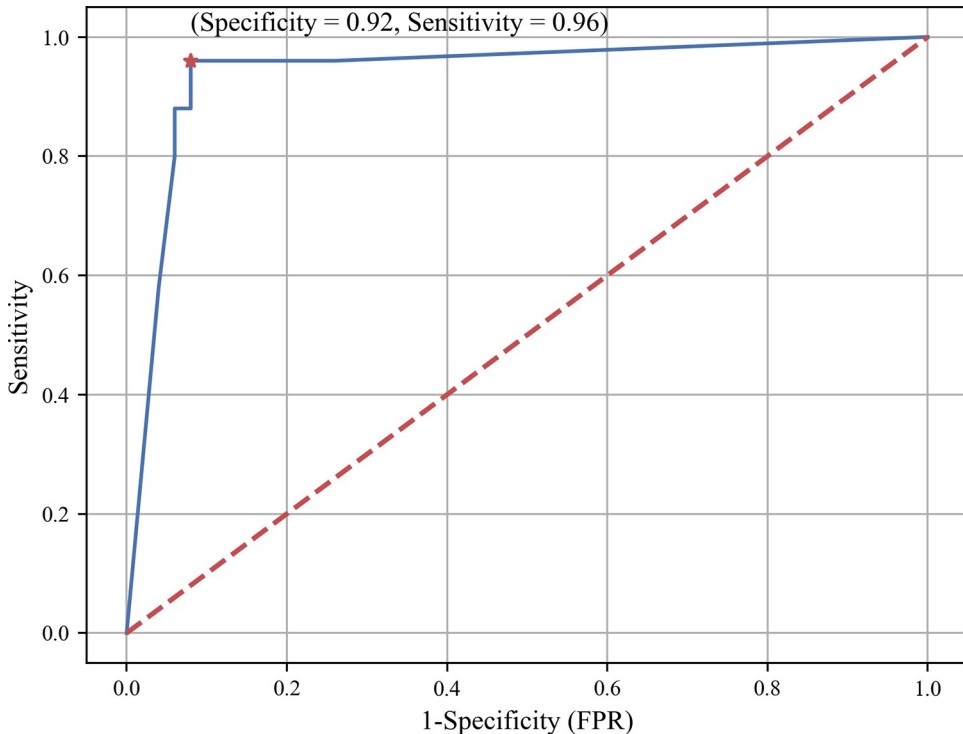

**Fig 8. ROC curve of the LDA classifier.**

results that significant improvements in classification are obtained based on these new features across different classification methods when compared with those based on the raw features. Similar to many other medical studies where the sample size of normal controls is significantly greater than that of patients, creating so-called sample imbalance problem that can negatively affect many conventional classification techniques. In this work, we found that SMOTE is an effective and easy to implement technique to address this issue. However, cautions must be taken to ensure that the synthesized samples mimic the true minority samples in terms of feature properties (e.g., n-gram counts can only take non-negative integers). In addition, to avoid overfitting, the synthetic samples should not be used as test samples. To improve classification performance and reduce the risk of overfitting, as well as to reduce model complexity, in this work we propose a two-step feature selection procedure. In the first step, the highly correlated and redundant features are removed through the evaluation of their Pearson correlation coefficients. In the second step, RFECV is utilized to further reduce the number of features. Through this two-step procedure, it is found that only ten features are needed to obtain the optimal cross-validation accuracy. Based on the raw features and the selected ten features, a systematic study and comparison of different linear and nonlinear machine learning classification techniques is conducted. It is found that with raw features, all classification methods, linear or nonlinear, fail to achieve high classification performance. In comparison, with the ten selected features, which contain nine features proposed in this work, the performances of all classification methods are significantly improved, indicating that the proposed features are more effective for characterizing speech disorder using speech LMs.

It is worth noting that small sample size has always been a limitation in biomedical studies due to labor, time, and other constraints. However, with careful separation of model training (including feature selection and hyperparameter tuning) and testing (using samples

completely left out of the model training process), significant conclusions can be drawn from analyses based on small number of samples. In this regard, MCVT is a robust technique for comparing different modeling techniques and assessing their performances with small number of test samples. Feature selection is also an effective way to avoid overfitting and reduce test variance, especially when there are more features than observations as in this study. Finally, SMOTE can help alleviate the problem by generating synthetic samples. A word of caution is that the above procedures ought to be limited to the training process on the training samples only to avoid overfitting by feature selection using test samples and bias introduced by the artificial samples.

## Author Contributions

**Conceptualization:** Marisha Speights Atkins, Q. Peter He.

**Formal analysis:** Kerul Suthar, Farnaz Yousefi Zowj, Q. Peter He.

**Methodology:** Kerul Suthar, Farnaz Yousefi Zowj, Marisha Speights Atkins, Q. Peter He.

**Project administration:** Marisha Speights Atkins, Q. Peter He.

**Resources:** Marisha Speights Atkins.

**Software:** Kerul Suthar, Farnaz Yousefi Zowj.

**Supervision:** Marisha Speights Atkins, Q. Peter He.

**Validation:** Kerul Suthar, Farnaz Yousefi Zowj.

**Visualization:** Kerul Suthar, Farnaz Yousefi Zowj.

**Writing – original draft:** Kerul Suthar, Farnaz Yousefi Zowj, Q. Peter He.

**Writing – review & editing:** Marisha Speights Atkins, Q. Peter He.

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
