## [Decision Letter · Decision Letter 0]

22 Feb 2022

PDIG-D-22-00004

Feature Engineering and Machine Learning for Computer-assisted Screening of Children with Speech Disorders

PLOS Digital Health

Dear Dr. He,

Thank you for submitting your manuscript to PLOS Digital Health. After careful consideration, we feel that it has merit but does not fully meet PLOS Digital Health's publication criteria as it currently stands. Therefore, we invite you to submit a revised version of the manuscript that addresses the points raised during the review process.

We look forward to receiving your revised manuscript.

Kind regards,

Henry Horng-Shing Lu

Section Editor

PLOS Digital Health

Journal Requirements:

1. Please amend your Financial Disclosure statement. If you did not receive any funding for this study, please simply state: “The authors received no specific funding for this work.”

2. Please update your Competing Interests statement. If you have no competing interests to declare, please state: “The authors have declared that no competing interests exist.”

3. Please provide a complete Data Availability Statement in the submission form, ensuring you include all necessary access information or a reason for why you are unable to make your data freely accessible. Note that it is not acceptable for the authors to be the sole named individuals responsible for ensuring data access.

PLOS defines a study's minimal data set as the underlying data used to reach the conclusions drawn in the manuscript and any additional data required to replicate the reported study findings in their entirety. Any potentially identifying patient information must be fully anonymized. 

If your research concerns only data provided within your submission, please write “All data are in the manuscript and/or supporting information files.” as your Data Availability Statement.

4. Please provide separate figure files in .tif or .eps format only and remove any figures embedded in your manuscript file. Please ensure that all files are under our size limit of 20MB.

For more information about how to convert your figure files please see our guidelines: https://journals.plos.org/digitalhealth/s/figures

Additional Editor Comments (if provided):

Reviewers' comments:

Reviewer's Responses to Questions

**Comments to the Author**

1. Does this manuscript meet PLOS Digital Health’s publication criteria? Is the manuscript technically sound, and do the data support the conclusions? The manuscript must describe methodologically and ethically rigorous research with conclusions that are appropriately drawn based on the data presented.

Reviewer #1: Yes

2. Has the statistical analysis been performed appropriately and rigorously?

Reviewer #1: No

3. Have the authors made all data underlying the findings in their manuscript fully available (please refer to the Data Availability Statement at the start of the manuscript PDF file)?

Reviewer #1: No

4. Is the manuscript presented in an intelligible fashion and written in standard English?

Reviewer #1: Yes

5. Review Comments to the Author

Reviewer #1: The authors proposed a computer-assisted approach for children speech disorders screening. They used the SpeechMark software to extract speech landmarks as features, based on which some standard feature engineering strategies (e.g., n-gram) and feature selection methods were applied. They also employed SMOTE method to handle the class imbalance issue. Standard machine learning algorithms (e.g., LDA, SVM, XGBoost, RF) were then trained based on those engineered features. The methods in this work are generally sound and they showed significant improvement in model performance with their feature engineering and selection strategies. However, there are a few concerns as listed below.

1. It is not clear if the authors had a separate testing set. It seemed that they used cross-validation or leave-one-out methods for feature selection, handled sample imbalance, and tuned hyper-parameters, all on the whole dataset. If they did not have a testing set separated before employing all the engineering and tuning, the models would likely be overfitted. This does not show in current results, but will show when applied to new data that the models have not seen. The authors should at least try to separate a testing set of about 20% of the data, do all the feature engineering and training on the rest, and test the models on the testing set.

2. Another shortcoming of this work is the limited data size – only from 51 children, which makes it hard to draw any significant conclusions. 

3. A related work using SpeechMark for landmarks extraction and machine learning algorithms for depression detection should be cited and compared. https://ieeexplore-ieee-org.ezproxy.lib.nctu.edu.tw/abstract/document/8682916

A few minor points:

1. There are duplicated references: 18 and 21, 19 and 22.

2. Figure 8 is not a standard ROC curve. Please refer to Figure 4 and link the left-most point to the origin and the right-most point to the top-right corner. Do not plot with circles. May consider labelling data values so the readers can see clearly the key points.

6. PLOS authors have the option to publish the peer review history of their article (what does this mean?). If published, this will include your full peer review and any attached files.

**Do you want your identity to be public for this peer review?** For information about this choice, including consent withdrawal, please see our Privacy Policy.

Reviewer #1: No

---

## [Decision Letter · Decision Letter 1]

8 Apr 2022

Feature Engineering and Machine Learning for Computer-assisted Screening of Children with Speech Disorders

PDIG-D-22-00004R1

Dear Prof. He,

We are pleased to inform you that your manuscript 'Feature Engineering and Machine Learning for Computer-assisted Screening of Children with Speech Disorders' has been provisionally accepted for publication in PLOS Digital Health.

Best regards,

Henry Horng-Shing Lu

Section Editor

PLOS Digital Health

Reviewer Comments (if any, and for reference):

Reviewer's Responses to Questions

**Comments to the Author**

1. If the authors have adequately addressed your comments raised in a previous round of review and you feel that this manuscript is now acceptable for publication, you may indicate that here to bypass the “Comments to the Author” section, enter your conflict of interest statement in the “Confidential to Editor” section, and submit your "Accept" recommendation.

Reviewer #1: All comments have been addressed

2. Does this manuscript meet PLOS Digital Health’s publication criteria? Is the manuscript technically sound, and do the data support the conclusions? The manuscript must describe methodologically and ethically rigorous research with conclusions that are appropriately drawn based on the data presented.

Reviewer #1: Yes

3. Has the statistical analysis been performed appropriately and rigorously?

Reviewer #1: Yes

4. Have the authors made all data underlying the findings in their manuscript fully available (please refer to the Data Availability Statement at the start of the manuscript PDF file)?

Reviewer #1: No

5. Is the manuscript presented in an intelligible fashion and written in standard English?

Reviewer #1: Yes

6. Review Comments to the Author

Reviewer #1: The authors have addressed all the issues raised in the previous review to satisfaction. For a minor suggestion, the authors are encouraged to check the tense in reporting their design, results, etc. - a past tense should be used.

7. PLOS authors have the option to publish the peer review history of their article (what does this mean?). If published, this will include your full peer review and any attached files.

**Do you want your identity to be public for this peer review?** For information about this choice, including consent withdrawal, please see our Privacy Policy.

Reviewer #1: No
